# Individual differences in bottom-up and top-down emotion generation

**Nadia Kako**[1]*, **Michelle Rozenman**[1], **Denis Dumas**[2], **Kateri McRae**[1]

**1** Department of Psychology, University of Denver, Denver, Colorado, United States of America,
**2** Department of Educational Psychology, University of Georgia, Athens, Georgia, United States of America

* Nadia.Kako@du.edu

## Abstract

Research has demonstrated there are two primary ways emotions can be generated. These include emotions generated from perception of simple physical properties of stimuli in the environment, known as "bottom-up" generated emotions, and emotions generated from cognitive appraisals of a cue or situation, known as "top-down" generated emotions. Experimentally, it has been shown these two forms of emotion generation recruit distinct neural and psychological processes. However, the extent to which individuals differ in how their emotions are generated, and how this variation may relate to other relevant constructs, has yet to be determined. Previous research has used self-report measures to assess individual differences in emotional responding. However, no measures to date have investigated individual differences in the degree to which people respond to bottom-up and top-down generated emotions specifically. A novel self-report questionnaire was developed to assess individual differences in bottom-up and top-down generated emotions. We report on three independent studies that evaluate the measures' psychometric properties and exploratory factor structure (Study 1; N = 149), confirmatory factor structure, associations of the measure with relevant psychopathology symptom domains, relevant established measures of emotional processes, and an established experimental task of emotion regulation (Studies 2 N = 230 and 3 N = 238). Factor analyses consistently indicated a two-factor structure for the items (Comparative Fit Index range = .94-.96) with adequate reliabilities. Correlation analyses revealed both top-down and bottom-up generated emotions were positively associated with internalizing symptoms and other measures relevant to emotion processing (significant *r* ranges:.15-.55). The relationships with extant measures indicated that we developed a measure that characterizes the use of cognition in the generation of negative emotion, as opposed to the use of cognition to control emotion. Future work should examine how this measure may be useful for affective and clinical science, including mechanistic treatment targets.

**Data availability statement:** Data is available on the OSF website: https://osf.io/w7h43/files/2pzsn.

**Funding:** Financial support was awarded to KM via the NSF Career Grant Number 1554683 (https://www.nsf.gov/funding/opportunities/career-faculty-early-career-development-program). The funders had no role in study design, data collection and analysis, decision to publish, or preparation of the manuscript.

**Competing interests:** The authors have declared that no competing interests exist.

## Introduction

Research has shown that the way an emotion is generated influences the subsequent emotional response [1,2]. Specifically, there is evidence that emotions can be generated through two distinct processes: either from the "bottom-up", driven by the perception of physical properties of one's environment, or from the "top-down," driven by cognitions and evaluations of a situation [3,4]. While there is experimental evidence indicating that how an emotion is generated leads to different profiles of emotional responding [5], less has been done to capture individual variation in these responses via self-report, and how this variation may be relevant to other important constructs. Researchers have had success in capturing individual differences in components of emotional responding via self-report measures. For example, researchers have successfully examined trait-based differences in emotional reactivity, including the intensity and duration of an emotional response [6]. However, few of these measures have specifically and separately examined differences in bottom-up and top-down generated emotions. While these established measures of reactivity are helpful, they don't delineate if the reactivity one is experiencing is (at least partially) separable based on the way the emotion that has elicited the response was generated.

### Bottom-up emotion generation

Bottom-up emotions are involved in responding to low-level perceptual cues, such as physical properties of an image or something in one's environment that reliably and directly elicits emotional responses. Bottom-up generated responses are thought to be "biologically prepared," and related to responses to threats present for much of evolutionary history [7,8]. For example, seeing a rattlesnake on a hiking trail would be a stimulus which elicits a threat response, including fear, from the bottom-up. Other examples of cues which elicit bottom-up generated emotions include fearful faces, threatening sounds or objects, and heights [9–11]. Responding quickly to bottom-up cues may help in immediately attending to emotional and possibly dangerous stimuli in one's environment, reflecting that bottom-up emotion generation is stimulus-dependent. Thus, individual differences in bottom-up generated emotions may reflect sensitivity to environmental stimuli or cues.

### Top-down emotion generation

In contrast to the perceptual nature of bottom-up generated emotions, top-down generated emotions are not directly dependent on the perception of certain environmental stimuli and are largely elicited by the (commonly conscious) appraisal of a situation [3,4]. The letters that comprise a text message are not commonly considered inherently emotional, however reading a message and consequently applying meaning to the *content* of the message can drive an emotional response [12]. For example, if one receives a message from a romantic partner that is short or curt, the message could be interpreted in such a way that indicates conflict in the relationship (e.g., "We need to talk"), eliciting a top-down generated emotional response.

Top-down emotions are predominantly conceptual and require individual interpretation or evaluation of one or more cues [4,13]. Interpretations leading to top-down emotions are flexible and largely contingent upon the cognitions and personal goals of the individual responding. Thus, individual differences in responding to top-down emotions may be related to an individual's unique temperament, well-being and/or mental health symptoms, life experiences and motivations.

Often bottom-up and top-down generated emotions work in concert throughout our daily lives. Indeed, individuals are regularly inundated with bottom-up stimuli and cues that are interpreted from the top-down simultaneously. Despite this complex overlap, experimental paradigms have been able to measure separable processes associated with these two types of emotion generation. In laboratory experiments, bottom-up emotions are commonly elicited through visual stimuli such as images thought to have low-level features that are emotional [14,15]. For example, in a study examining fear learning, participants were asked to perform a task involving fearful faces where they either completed the task immediately, observed others perform the task first, or received verbal instruction about the task and then completed it. The task found evidence for two dissociable processes of emotional learning; one which relied primarily on exposure to perceptual cues (the experiential and observation conditions) and one which relied more heavily on explicit linguistic instruction (the instruction condition [14].

The use of perceptual stimuli in experiments studying bottom-up emotions has consistently demonstrated activation in the amygdala [10]. For example, in a study evaluating differences in neural representation of bottom-up and top-down generated emotions, participants showed consistent amygdala and visual cortex activation when viewing negatively valenced images [5]. Amygdala activation likely occurs due to its role in drawing attention to emotionally provocative stimuli in one's environment [15,16], allowing for rapid threat detection.

In contrast, top-down generated emotions in the laboratory are commonly elicited via language, either through verbal instruction or reading vignettes or captions [3,4]. For example, one study used a task in which pictures were paired with either congruent or incongruent captions. For example, a congruent trial would include an image of a staircase with the caption "where she broke her neck" while an incongruent trial would include the staircase image coupled with the caption "he is being exploited". Participants rated their affect as significantly more negative for congruent blocks, suggesting the higher-level, cognitive nature of the task was mediating emotional responses [4].

This study demonstrated top-down generated emotions elicit specific psychological processes and specific underlying neural mechanisms. The study found that the medial prefrontal cortex (mPFC) was the primary region engaged during emotional responses to congruent picture-caption pairs [4]. Other experiments that have utilized linguistic cues to study top-down processing have consistently shown activation in mPFC. mPFC is implicated in processing top-down generated emotions due to its role in ascribing meaning to potentially affective cues by integrating affective and cognitive inputs [5]. Further, mPFC is recruited when attempting to reevaluate an image to either increase or decrease its negative valence [17].

In sum, bottom-up generated emotions can be defined by their more direct reliance on perception of environmental cues. Across contexts, a dog growling and baring its teeth elicits a relatively immediate fear response. In contrast, top-down generated emotions do not require encounters with physical stimuli to elicit an emotional response, and instead rely on activation of concepts, communicated through symbolic meaning such as language, and are modified by context. The current literature indicates there is evidence that bottom-up and top-down emotion generation occurs via somewhat distinct neural processes and therefore may reflect distinct pathways for an individual's emotion generation in daily life

## Current individual difference measures related to bottom-up and top-down emotion generation

Although there is evidence bottom-up and top-down generated emotions are governed by separable processes, there are no individual difference measures that specifically or separately address them. Current measures may indirectly address these processes through personality and clinical indicators which measure aspects of emotion processing.

Common individual difference measures relevant to emotional responding have focused on different components of an emotional response such as the frequency, magnitude, and duration of the emotion, such as Nock and colleagues' Emotional Reactivity Scale [6]. While helpful, this measure does not directly examine variation in bottom-up and top-down emotion reactivity specifically. In addition, different personality measures characterize differences related to how one responds to cues in one's environment (e.g., responding to loud noises; [18] and one's internal reactions and cognitions to emotions [19]). Some measures have also attempted to examine how individuals cope or regulate their emotional responses, examining cognitive and behavioral strategies [20]. One extant measure attempts to characterize the importance of bottom-up and top-down emotions in the context of emotional reactivity and regulation. The Cognitive Mediation Beliefs Questionnaire (CMBQ; [21] assesses the degree to which one believes emotions are generated from the top-down or bottom-up. The two scales most rigorously measure top-down regulation and bottom-up generation beliefs, so do not fully characterize and contrast bottom-up and top-down emotion generation.

While each of these approaches have provided important insight into emotional processes and how they interact with other systems, they do not differentiate between bottom-up and top-down generated emotions, and thus it is unclear whether these emotions are even separable via self-report. It is also unclear whether variation in these emotions might relate to symptoms of psychopathology. Previous individual difference research suggests that personality and other trait-based aspects of emotional responding are related to psychopathology symptoms [6,22]. This suggests that better understanding bottom-up and top-down emotions could provide additional information about factors that may contribute to the development and maintenance of psychopathology.

It is not known whether relationships between emotional responding and psychopathology are specific to different types of self-reported emotion generation or span across both stimulus-driven (bottom-up) and cognitive (top-down) ways of generating emotion. In the future, we believe that a self-report measure specifically developed to assess both of these emotion generation mechanisms will be helpful in this important line of research and help clarify how emotion generation occurs in the context of, and serves as risk for onset and maintenance of, internalizing problems.

## The present study

The current study aimed to determine whether it was possible to generate self-report items that reflected two separable emotion generation processes, those more closely related to cognitions/thoughts, and those related to perception/environment. If we were able to develop such items, we then wanted to assess whether two subscales were an adequate fit for the generated items; these two subscales may indicate novel separability of bottom-up and top-down generated emotions in a self-reported questionnaire that might be easily and quickly administered to research participants and individuals receiving treatment for internalizing problems alike.

Our work in developing and testing psychometric properties of such a questionnaire was guided by several *a priori* principles. First, we did not conceptualize individual differences in bottom-up and top-down emotions as personality traits, nor were we certain about how variation in one may relate to the other. However, we did expect differences in generation of both types of emotion to possibly contribute to emotionally relevant constructs, such as psychopathology. Thus, assuming we had support for our first and second goals, the third goal was to identify how this measure (and its possible subscales) may associate with symptoms of internalizing psychopathology (i.e., depression and anxiety). Lastly, the measure was examined for its convergent and divergent validity to other extant emotion measures, including established questionnaires and task-based measures of emotion regulation.

We report three primary studies which follow guidelines for the development and validation of psychometric measures [23]. The first study reports on the item development and refinement of the item pool, the scale development, and the underlying structure of the final scale items. The preregistered second study, which utilized a novel sample, reports on scale evaluation, the measure's relationship to symptoms of internalizing psychopathology, and other measures of emotion. Finally, the third study used another novel sample to report on scale evaluation and the measure's relationship to

other measures of internalizing psychopathology, and self-report measures of emotion and emotion regulation via survey and experimental task.

## Study 1: Development and Factor Structure of the Measure

### Materials and methods

The items developed were hypothesized to have reasonable internal consistency with acceptable to strong reliability levels. Because the experimental literature indicates that bottom-up and top-down generated emotions are separable, but work in tandem, the two hypothesized factors were expected to possibly be positively correlated. However, because the neural evidence also demonstrates these are two somewhat distinct processes, it was predicted that the association would be relatively weak.

### Item development

To generate the initial pool of items, the experimental literature on bottom-up and top-down emotions was first reviewed, as well as concepts from the construal level literature [24]. Current individual difference measures that may partially reflect aspects of top-down or bottom-up emotions were then evaluated, including measures of cognition [19,25], sensitivity to stimuli in one's environment [18], emotion reactivity [6], and emotion regulation [20]. This process resulted in the identification of specific wording to consider for questions and aided in the development of items. The goal was to generate more items than may be needed to run exploratory factor analyses (EFA) that would determine the best fitting item distribution and maximize internal consistency.

### Feedback from experts and item refinement

Items were then presented to five experts within related domains including psychometrics ($N=1$), clinical psychology ($N=2$), and affective science ($N=2$). Experts provided feedback on the items both for conceptual validity and ease of understanding. Feedback was also elicited during multiple lab presentations from undergraduate and graduate students who were not familiar with the study. Feedback was then incorporated to refine the items for clarity. A total of 41 items were included (see supplementary material for more information about how items were reduced for the EFA described below). Items were presented on a sliding scale with three anchors (1=strongly disagree, 4= neither agree or disagree and 7=strongly agree), with possible scores ranging from 1-7 per item. Data on these items were collected during a study outlined below.

### Participants

The data for this measure was collected in conjunction with a separate preregistered study (see [26] for detail of this separate study) which utilized an online platform to collect data. Thus, while our exploratory analyses were not preregistered, these data followed the same preregistered exclusions outlined by the study from which it was collected. A total of 193 undergraduates across North America were initially recruited through the online, anonymous survey platform, Prolific [27]. Eligibility criteria included current enrollment in an undergraduate degree program at a North American university and being at least 18 years old.

### Ethical approval and consent to participate

Participants participated in this study between August 4 and September 7, 2021, and were compensated approximately $10 per hour for participation. All participants completed written informed consent via the online survey platform, and all study procedures were approved by the Institutional Review Board of the University of Denver prior to the initiation of any study procedures.

Several exclusion criteria were applied to the dataset to ensure quality. Data were excluded from participants who exited the study survey early and therefore did not respond to all items ($N=14$). Data were also excluded if participants failed one or both attention check items included in the survey ($N=18$). Data were excluded from participants who completed the survey in under 20 minutes ($N=19$), as based on pilot testing, the survey was expected to take approximately 30–36 minutes to complete conscientiously. Lastly, data were excluded from participants with scores that were greater than three standard deviations from the mean of measures collected ($N=3$). The analyses and results below are based on a final sample of 149 participants. See Table 1 for demographics information of this sample.

## Procedure and questionnaires

Participants completed the study in a single-session online Qualtrics survey with a duration of approximately 40 minutes. Participants were first required to complete informed consent, and then completed a demographics questionnaire, a battery of psychometric questionnaires, including the bottom-up top-down (BUTD) measure, and two questionnaires unrelated to the current study. The BUTD measure items were presented in a random order.

## Data analysis

**Scale development.** 10 total items were subjected to factor analysis. The Kaiser-Meyer-Olkin (KMO) test and Bartlett's test of sphericity were used to assess sampling adequacy and whether the data were appropriate for factor analysis. The following indices were used to assess goodness of fit: The Root Mean Square Error of Approximation (RMSEA), Comparative Fit Index (CFI), Tucker-Lewis index (TLI), and the Standardized Root Mean Square Residual (SRMR). Specifications for this two-factor structure analysis included running a parallel analysis to examine suggested number of factors, forcing a two-factor solution, suppressing coefficients less than.2, maximum likelihood estimation and an oblique Promax rotation, as it was expected that the factors would be correlated. Analyses were implemented in R, version 4.3.0, and the lavaan package, version 0.6.15 [28].

**Table 1. Reported demographic characteristics of study 1 sample.**

| Age | 20.4±3.3 |
|---|---|
| Sex | |
| Female | 133 |
| Male | 16 |
| Gender | |
| Man | 17 |
| Non-binary | 3 |
| Woman | 129 |
| Race | |
| African American | 9 |
| American Indian or Alaskan Native | 0 |
| Asian | 5 |
| White | 122 |
| Multiracial | 7 |
| Native Hawaiian or Pacific Islander | 1 |
| Ethnicity | |
| Hispanic/Latinx | 21 |
| Not Hispanic/Latinx | 127 |

## Results

Both tests (KMO = .77) and Bartlett's test of sphericity, $X^2$ [29] = 405, $p < .001$, indicated the data were suitable for analysis. The scree plot of eigenvalues from the parallel analysis indicated two factors with values of at least 1 (Fig 1).

The forced-factor factor analysis indicated two factors, accounting for more than 48% of the variance in the data. The first factor revealed a top-down factor containing six items and the second factor revealed a bottom-up factor containing four items. The two factors were positively correlated ($r = .39$). The two scales had reasonable internal consistency with alpha of .81 for the top-down scale and .68 for the bottom-up scale. McDonald's omega was also calculated as an additional measure of reliability given the limitations of Cronbach's alpha (see [30] for more information on use cases for alpha, omega, and coefficient H), with comparable results (top-down =.81, bottom-up =.68). The results indicated that a two-factor approach measured distinct bottom-up and top-down emotion generation factors and that both scales had reasonable internal consistency. Table 2 outlines the results of the EFA.

The forced two-factor correlated model produced the following fit indices, $\chi^2 = 45.07$, df = 26, $p = .01$, RMSEA = .07 (90% CI = .04-.11), CFI = .94, TLI = .90, SRMR = .05, all of which indicated reasonable fit (Table 3). See S2 Table in supporting information for factor loadings.

## Study 2: Confirmatory factor structure and relationship to extant measures

The first study demonstrated that a 10-item novel self-report measure of emotion generation reflected two factors: top-down and bottom-up. A separate, independent sample was recruited to determine whether we could replicate the two-factor structure produced from the first study. In this second study, we also further assessed the measure's fit by statistically comparing the two-factor structure with another factor structure to determine which would better capture variability observed in the data. Furthermore, we predicted that both factors would be positively correlated with extant individual difference measures of internalizing symptoms and other measures of emotion potentially related to bottom-up and top-down generated emotions. More specifically, we believed it was possible the bottom-up scale would be more strongly correlated with symptoms of anxiety, given bottom-up generated emotions involve perception, and perceptual awareness

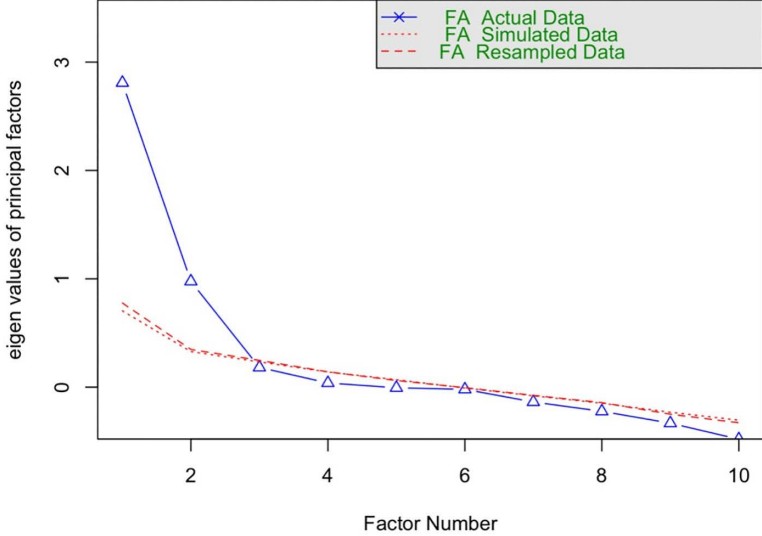

**Fig 1. Scree Plots Results from Parallel Analysis.**

**Table 2. EFA outcomes for the two-factor model with factor loadings.**

| Item | Factor Loading | Communalities Extraction | % Variance | α | w | Eigen Value | Factor correlation |
|---|---|---|---|---|---|---|---|
| **Factor 1: Top-down** | | | 34.2 | .81 | .81 | 3.44 | .39 |
| My thoughts tend to cause some sort of emotional response in me. | .82 | .65 | | | | | |
| When I'm feeling emotional, it's often because of thoughts I've had. | .73 | .50 | | | | | |
| When I am feeling emotional it's often because of how I have interpreted a situation. | .44 | .35 | | | | | |
| My thoughts often affect how I feel. | .41 | .29 | | | | | |
| How I interpret conversations tends to affect my emotions. | .80 | .56 | | | | | |
| When I experience negative emotions, I tend to get caught up in my thoughts. | .71 | .45 | | | | | |
| **Factor 2: Bottom-up** | | | | | | | |
| Viewing emotional images tend to cause an emotional response in me. | .44 | .20 | 14.2 | .68 | .68 | 1.83 | .39 |
| My emotions often arise from experiencing things in my environment (e.g., sounds, images, or scenes). | .59 | .36 | | | | | |
| My emotions come from physical properties of a situation such as the scenery and sounds of a location. | .80 | .54 | | | | | |
| I tend to react to displays of emotions by others. | .40 | .26 | | | | | |

*Note.* Total variance explained = 48.4%, $X^2$ = 45.07, *p* =.01.

**Table 3. Fit Indices of the Two-Factor Solution.**

| $\chi^2$ | df | CFI | TLI | RMSEA | SRMR |
|---|---|---|---|---|---|
| 45.07 | 26 | .94 | .90 | .07 | .05 |

*Note.* All models were statistically significant at the *p* < .05 level. $\chi^2$ = chi-square test statistic, RMSEA = Root Mean Square Error of Approximation, CI = Confidence Interval, CFI = Comparative Fit Index, TLI = Tucker-Lewis Index.

to possibly threatening stimuli is commonly heightened in anxiety [31]. In contrast, we predicted the top-down scale would be more strongly related to symptoms of depression, given that top-down generated emotion involve cognition, and ruminative thinking is commonly involved in depression [32]. This study was preregistered on the Open Science Framework website (https://osf.io/3hm48).

To address our fourth outcome, examining how the BUTD measure related to extant measures of emotional responding, measures that assessed specific aspects of cognition, sensory sensitivity, and emotion regulation were included. Like the predictions with internalizing measures, we hypothesized that the bottom-up and top-down factors could have both distinct and overlapping relationships with the measures collected. These measures varied in their specificity; with some assessing solely aspects of cognition or sensory sensitivity, some that contained subscales that measured both, and some that examined more basic aspects of an emotional response, including emotional reactivity. Generally, it was hypothesized that measures that assess aspects of cognition including both adaptive and maladaptive forms of cognition (e.g., rumination) would be positively related to the top-down scale, and measures that assess sensitivity to external stimuli would be positively related to the bottom-up scale. The emotion regulation measures were included to explore how the newly developed measure may be related to other relevant outcomes of well-being and emotional responding (see S1 Table in

supporting information for specific predictions for all measures included), and thus we did not have specific hypotheses outlined.

Of note, while the literature supports that bottom-up and top-down generated emotions are two separable processes, both these types of emotions often work together. Thus, individuals could respond in a way that indicates that they are highly responsive (or not) to both bottom-up and top-down generated emotions.

## Methods

**Participants.** Participants in Study 2 were undergraduates at the University of Denver recruited through a participant pool managed by the Department of Psychology. The same eligibility criteria and informed consent processes were applied as in Study 1 (e.g., enrollment in an undergraduate institution in the United States, being at least 18 years old).

**Ethical approval and consent to participate.** Participants completed the study during the students' Autumn and Winter Quarters of the 2021–2022 academic year between October 5th, 2021 and February 14th, 2022. Students received course credit for their participation. Continuing with guidelines previously outlined [23], the goal was to collect at least a 10:1 ratio of usable participants to items. The final sample ($N=230$) met this goal.

A total of 274 participants were initially recruited. Similar exclusion criteria were applied to this sample as in Study 1, including completing the study in under 10 minutes, failing any attention checks ($N=24$), scores greater than three standard deviations from the mean on any measure ($N=15$) and the addition of invariant responding (e.g., straight line responding with no change in answer selection; $N=3$), resulting in a final sample of 230 participants. See Table 4 for demographics information of the sample.

## Procedure

Participants completed a battery of self-report measures via Qualtrics which contained the 10-item bottom-up and top-down retained from Study 1, as well as a measure of basic demographics and the self-report measures related to emotion processing listed below (see Measure section). To control order effects, a pseudo-random order for the measurement

**Table 4. Reported demographic characteristics of study 2 sample.**

| Age | 19.3±1.35 |
|---|---|
| Sex | |
| Female | 192 |
| Male | 38 |
| Gender | |
| Man | 37 |
| Non-binary | 3 |
| Woman | 187 |
| Race | |
| African American | 8 |
| American Indian or Alaskan Native | 2 |
| Asian | 21 |
| White | 176 |
| Multiracial | 12 |
| Native Hawaiian or Pacific Islander | 1 |
| Ethnicity | |
| Hispanic/Latinx | 22 |
| Not Hispanic/Latinx | 198 |

battery was used with higher priority measures presented in the first half of the battery. Items from the BUTD measure were presented in a random order. All participants completed written informed consent via an online survey and the procedures for this study and were approved by the University of Denver's Institutional Review Board prior to initiation of any study procedures.

## Measures

**Internalizing symptom measures. Beck depression inventory (BDI-II).** To assess the BUTD measure's relationships with various mental health symptoms, we included a measure of depressive symptoms. The Beck Depression Inventory is a 21-item self-report inventory that assesses symptoms of depression [33]. Sample items include selecting one out of multiple options about a symptom such as: "I do not feel sad.; I feel sad.; I am sad all the time and I can't snap out of it; I am so sad and unhappy that I can't stand it." Higher scores indicate greater depression symptoms. The current sample demonstrated strong reliability, with w = .90 and α = .90, which is slightly stronger than previous studies which demonstrated alphas of.86 for clinical populations, and.81 for non-clinical populations [33].

**State-trait anxiety inventory –trait (STAI-T) scale.** To assess the BUTD measure's relationships with various mental health symptoms, we included a measure of anxiety symptoms. The STAI is a 40-item measure of symptoms of anxiety that assess for both state (STAI-T; 20 items) and trait (STAI-T; 20 items) anxiety symptoms [34]. Given the primary interest of this measure was symptoms of anxiety rather than state-level stress, this study focuses on the Trait sub-scale. STAI-T items include: "I worry too much over something that really doesn't matter". All items are rated on a 4-point scale. Higher scores indicate greater anxiety symptoms. The current sample demonstrated acceptable reliability: STAI-T α = .92, w = .92, STAI-S α = .92, w = .91 which was comparable to previous evaluations of the scales by the original creator. STAI-T = .90, STAI-S = .93 [34,35].

**Emotion measures. Emotion reactivity scale (ERS).** To assess the BUTD measure's relationships with broader emotional processes, we included a measure of emotion reactivity. The ERS is a 21-item self-report scale developed to assess an individual's emotion reactivity [6]. It contains three subscales measuring emotion Sensitivity, Intensity, and Persistence. Along with its three subscales, it captures a single factor (Emotion Reactivity) which yielded strong internal consistency in the current sample (α = .94, w = .94), consistent with its original evaluation of internal consistency (α = .94; [6]. Sample items such as: "When I'm emotionally upset, my whole body gets physically upset as well", and "When something happens that upsets me, it's all I can think about for a long time" reflect items in the intensity and persistence subscales respectively.

**The highly sensitive person scale (HSPS).** To assess whether the bottom-up scale specifically related to heightened perceptual awareness, we included a measure that assesses sensitivity to environmental stimuli. The HSPS is a 27-item scale which measures sensory processing sensitivity [18]. The measure enquires about sensitivity to environmental stimuli such as "Are you easily overwhelmed by things like bright lights, strong smells, coarse fabrics, or sirens close by?" the items measure a single factor known as high sensory processing sensitivity and demonstrated strong internal consistency (α = .88, w = .87) in the current sample, similar to previous evaluations of consistency demonstrated by the original authors (α = .87; [18].

**The dimensions of openness to emotions (DOE).** To examine the measure's convergence with specific emotional processes, we included the DOE. The DOE scale is a 20-item measure used to assess multiple dimensions of affect processing at the trait level for individuals [36]. The DOE consists of five subscales that assess various forms of affect processing including Perception of Internal Bodily Indicators of Emotions, Perception of External Bodily Indicators of Emotions, Regulation of Emotions, Communication and Expression of Emotions, and Cognitive Conceptual Representation of Emotions. Sample items include: "I know exactly what emotional state I am in", "My strong feelings are accompanied by internal bodily reactions". Reliability for the five sub-scales is satisfactory with alphas and omegas respectively of: Cognitive Conceptual Representation of Emotions α = .85, w = .86, Communication and Expression of Emotions α = .84, w = .84,

Perception of Internal Bodily Indicators of Emotions α = .78, w = .77, Perception of External Bodily Indicators of Emotions α = .77, w = .78 Regulation of Emotions α = .63, w = .57 from the current sample, which are comparable with the original internal consistency of each scale reported by the author (α ranges = .67-.83; [36]).

**Cognitive mediation beliefs questionnaire (CMBQ).** To examine the measure's construct validity, we included the CMBQ which assesses constructs more closely related to bottom-up and TD processes. The CMBQ is a 15-item scale that measures beliefs about emotions and whether they are cognitively mediated [21]. The CMBQ contains two subscales, Cognitive Mediation change (C-M), which measures the degree to which people believe emotions are cognitively meditated, and Stimulus-Response generation (S-R), which measures the degree to which people believe emotions arise from responding to external stimuli. Sample items for the C-M include: "to change how I feel, my thoughts about the situation need to change", and for S-R: "My feelings are entirely determined by people's actions towards me". The sub-scales demonstrated strong reliabilities, both with alphas of.90 and omegas of.88 for the C-M subscale and.90 for the S-R subscale for the current sample. The current sample demonstrated slightly stronger internal consistency for the C-M scale, with a previous reported α = .81, and the same internal consistency for the S-R with a previous reported α = .90 [21].

**Need for cognition scale (NCS).** To assess whether the top-down scale specifically related to cognitive processes, we included several measures of cognition. The NCS is an 18-item scale that measures an individual's preference and enjoyment to engage in effortful cognitive evaluations. The scale measures a single factor (need for cognition) with strong reliability in the current sample (α and w = .87) that was comparable to the originally reported a α = .90 [37]. Example items include: "I find satisfaction in deliberating hard and for long hours" and "I would prefer complex to simple problems".

**The rumination reflection questionnaire (RRQ).** The RRQ is a 24-item scale that distinguishes between ruminative and reflective thoughts [19]. It is composed of two factors; Rumination and Reflection, with items such as "My attention is often focused on aspects of myself I wish I would stop thinking about" to identify ruminative thoughts and "I love analyzing why I do things" to categorize reflective thoughts. The RRQ demonstrated strong reliability (both αs = .92) in the current sample, closely aligned with previously reported alphas (Reflection = .91, w = .92; Rumination α = .90, w = .91; [19].

**Cognitive flexibility inventory (CFI).** The CFI is a 20-item scale that measures one's ability to replace maladaptive thoughts with more balanced thinking [25]. The CFI includes two subscales, Control, which measures the tendency to perceive a sense of control over difficult situations and Alternative, the ability to produce alternative explanations or solutions to life events or difficult situations. The Alternative and Control sub-scales demonstrated acceptable reliability with αs = .78 and.83 and ws of.80 and.86 in the current sample, the alphas were somewhat lower than previously reported by the original authors (Alternative α = .91, Control α = .84 [25]. Example items include "It is important to look at difficult situations from many angles." and "When I encounter difficult situations, I feel like I am losing control".

**Exploratory measures. The big-five inventory (BFI-V).** The BFI-V is a 44-item measure that assesses the big five personality dimensions: Extraversion, Agreeableness, Openness, Neuroticism, and Conscientiousness [38].There is particular interest in the Neuroticism dimension for this sample given its associations with emotion reactivity and well-being [39,40]. Participants are asked to identify how strongly they agree with statements related to the five dimensions including Neuroticism such as "Can be tense". The current sample demonstrated acceptable reliability (Neuroticism α and w = .86), consistent with previously reported reliability (α = .86; [38].

**Emotion regulation questionnaire (ERQ).** The ERQ is a 10-item measure that assesses two dimensions of emotion regulation which are the tendency to use cognitive reappraisal and expressive suppression [20]. Sample items include "When I want to feel more positive emotion (such as joy or amusement) I change what I'm thinking about" and "I keep emotions to myself" for Reappraisal and Suppression respectively. The current sample demonstrated adequate reliability: Reappraisal α = .85, w = .86, Suppression α = .76, w = .81 comparable to and in some cases stronger than previously reported reliabilities (Reappraisal ranges α = .72-.76, Suppression ranges α = .64-.69; [41]).

## Data analysis

**Scale evaluation.** Data were subjected to a CFA, using the same Structural Equation Modelling package lavaan in R as the first study. A two-factor correlated model and one-factor model were tested (two CFAs in total) using a robust maximum likelihood estimation. The different models were assessed to determine whether bottom-up and top-down emotions could be measured via two factors or a single factor, given that the factors in Study 1 were correlated. The two models were then compared to estimate which model offered the best fit of the data using a $\chi^2$ difference test to compare the two models. Fit indices for both models were also compared. Modification indices were run for both models to identify whether any alternate models provided a better fit of the data. Modification indices were only examined for large reductions in $\chi^2$ change value (e.g., greater than or equal to 10) and the alternate model aligned with the theoretical basis of bottom-up and top-down generated emotions [42,43].

**Extant measures.** Extant measures of emotional responding were computed in SPSS software version 25. Computations for each measure were done by following the guidelines outlined by the original creators, including reverse coding and calculating the total and sub-scale scores. The reliability of each measure was also evaluated using Cronbach's alpha and McDonald's omega (see Measures above for detailed reliability reporting). Bivariate correlations of each scale, and sub-scales when relevant, were then run with the factor scores extracted from the CFA to examine convergent and discriminant validity (see supplement for table of predicted correlations). Factor scores were used as the primary measurement of bottom-up and top-down generated emotions because factor scores consider the differential loading of individual items that contribute to a scale and thus reduce error introduced by simple summing, which assumes each item contributes equally to a scale score [44].

## Results

**Confirmatory factor analysis.** The 10-item one-factor model was not an acceptable fit, $\chi^2 = 111.29$, df = 35, $p < .001$, RMSEA = .11 (90% CI = .08-.013), CFI = .84, TLI = .80, SRMR = .08. The 10-item two-factor correlated model was an acceptable fit, $\chi^2 = 75.94$, df = 34, $p < .001$, RMSEA = .08 (90% CI = .06-.01), CFI = .91, TLI = .88, SRMR = .06. Upon the examination of the modification indices, error covariance between two of the bottom-up items warranted further inspection given it met the prespecified $\chi^2$ change criterion of being greater than or equal to 10 and appeared sensical and theoretically consistent with the data and the bottom-up and top-down generated emotion constructs. Specifically, the qualitative meaning of the items indicated a potential shared variance related to beliefs about how bottom-up emotions arise from external stimuli (e.g., beliefs that emotions arise from perceptual cues such as sights or sounds in the environment). Given this variance appeared sensical, the two-factor correlated model was then run with allowance of error covariance between the two bottom-up items and produced a good fit, $\chi^2 = 51.99$, df = 33, $p = .019$, RMSEA = .05 (90% CI = .02-.08), CFI = .96, TLI = .95, SRMR = .05, and then compared to the one factor model with the same allowance of error covariance: $\chi^2 = 81.28$, df = 34, $p < .001$, RMSEA = .08 (90% CI = .06-.11), CFI = .90, TLI = .87, SRMR = .06. The final model comparison indicated the two-factor correlated model with error-covariance was the best fit (see Table 5 for model comparison and fit indices and Table 6 for factor loadings and intercepts).

**Table 5. Fit indices of the two CFA models.**

| Model | $\chi^2$ | df | CFI | TLI | RMSEA | SRMR | $\Delta\chi^2$ |
|---|---|---|---|---|---|---|---|
| **1. Correlated two- factor** | **51.99** | **33** | **.96** | **.95** | **.05** | .05 | Model 1 vs Model 2: 29.29, df diff = 1, $p < .001$ |
| 2. One factor | 81.28 | 34 | .90 | .87 | .08 | .06 | |

*Note.* All models were statistically significant at the $p < .05$ level. The bolded model demonstrated the best fit of the two models.

**Table 6. Intercepts and loadings of each item in the CFA.**

| Top-Down Item | Bottom-Up Item | Intercept | Loading |
|---|---|---|---|
| My thoughts tend to cause some sort of emotional response in me. | | 5.67 | .83 |
| When I'm feeling emotional, it's often because of thoughts I've had. | | 5.56 | .73 |
| When I am feeling emotional it's often because of how I have interpreted a situation. | | 5.37 | .60 |
| My thoughts often affect how I feel. | | 5.95 | .76 |
| How I interpret conversations tends to affect my emotions. | | 5.59 | .56 |
| When I experience negative emotions, I tend to get caught up in my thoughts. | | 6.07 | .61 |
| | Viewing emotional images tends to cause an emotional response in me. | 4.73 | .82 |
| | My emotions often arise from experiencing things in my environment (e.g., sounds, images, or scenes). | 4.82 | .49 |
| | My emotions come from physical properties of a situation such as the scenery and sounds of a location. | 4.19 | .31 |
| | I tend to react to displays of emotions by others. | 5.09 | .92 |

*Note.* Questions were on a continuous, 7-point Likert scale where 1= strongly disagree, 4= neither agree or disagree and 7= strongly agree.

**Relationship to internalizing symptoms.** The bivariate correlations revealed that both the bottom-up and top-down factor scores were positively associated with internalizing symptoms (STAI-T: ($rs$(192) =.23,.32 $ps$ < .001; BDI: ($rs$(192) =.22,.37, $ps$ < .01)*,* suggesting a small transdiagnostic relationship with measures of depression and anxiety.

**Validity assessment and association with extant measures.** The Pearson's correlation analysis (Table 7) indicated both the bottom-up and top-down factors were positively associated with the ERS ($r$(193) =.55, $p$ < .001 and $r$(193)=.38, $p$ < .001 respectively), as predicted. The top-down factor overall was not positively related to measures of cognition related to flexibility and control as expected. However, it was positively related to the Rumination sub-scale of the RRQ ($r$(192) =.44, $p$ < .001) as predicted, and negatively associated with the CFI Control sub-scale ($r$(192) = -.25, $p$ < .001) and the Need for Cognition Scale ($r$(192) = -.15, $p$ < .001). This pattern of correlations suggests the top-down scale may be measuring cognitively generated emotion reactivity. As predicted, the bottom-up factor was positively associated with measures of reacting to external stimuli including the HSPS ($r$(193) =.38, $p$ < .001) and the S-R sub-scale of the CMBQ ($r$(190)=.24, $p$ = .001) as well as both DOE sub-scales that measure internal and external bodily cues of emotion ($rs$(191) =.28 and.21, $p$ < .001 respectively).

**Secondary measures analysis.** The analysis of secondary measures of interest revealed that the top-down factor was not associated with emotion regulation strategy use, and the bottom-up factor was negatively associated with the Expressive Suppression sub-scale of the ERQ ($r$(193) = -.15 $p$ = .032)*.* Both factors were also negatively associated with the Regulation of Emotions subscale of the DOE (top down: $r$(191) = -.41; bottom-up $r$(191) = -.33, $ps$ < .001). Lastly, both the bottom-up and top-down factors were associated with Neuroticism, a personality trait that has been associated with increased emotional reactivity ($rs$(193) =.41,.22, $ps$ < .01;40). For more information about how the BUTD measure performed in comparison to a measure of like constructs, please see S3 Table in the supporting information file.

Generally, it appeared that there was some separation between the bottom-up and top-down scales, however, both scales correlated with many of the same constructs, with only small descriptive differences in $r$ values. Both scales appeared to be related to psychopathology as predicted. Furthermore, it appeared that the top-down scale specifically seemed to be consistently and more strongly associated with measures of emotion reactivity generally. Furthermore, the

**Table 7. Bottom-up top-down correlations with extant measures.**

| | Top-Down | Bottom-Up |
|---|---|---|
| **Top-down Factor** | – | .66*** |
| Rumination Reflection Questionnaire – Rumination | **.44*** | .26* |
| Rumination Reflection Questionnaire – Reflection | -- | -- |
| Cognitive-Mediation Beliefs Questionnaire – Cognitive-Mediation | -- | -- |
| Cognitive Flexibility Inventory – Control | -.25*** | -- |
| Cognitive Flexibility Inventory – Alternative | -- | -- |
| Dimensions of Openness to Emotions – Communication and Expression of Emotions | -- | .16*** |
| Dimensions of Openness to Emotions – Cognitive Conceptual Representation of Emotions | -- | -- |
| Need for Cognition Scale | -.15* | -- |
| Emotion Reactivity Scale | **.55*** | **.38*** |
| **Bottom-Up Factor** | .66*** | – |
| Highly Sensitive Person Scale | .51*** | **.38*** |
| Cognitive-Mediation Beliefs Questionnaire – Stimulus-Response | .36*** | **.24*** |
| Dimensions of Openness to Emotions – Perception of External Bodily Indicators of Emotion | .18* | .21** |
| Dimensions of Openness to Emotions – Perception of Internal Bodily Indicators of Emotion | .29*** | .28*** |
| Big-Five Inventory – Neuroticism | **.41** | .22** |
| Emotion Regulation Questionnaire – Reappraisal | -- | -- |
| Emotion Regulation Questionnaire – Suppression | -- | **-.15*** |
| Dimensions of Openness to Emotions – Regulation of Emotions | -.41*** | -.33*** |
| Beck Depression Inventory Second Edition | **.37*** | .22** |
| State-Trait Anxiety Inventory – Trait | **.32*** | .23** |

*Note.* Bolded values indicate observed value is aligned with predicted value. * $p < .05$. ** $p < .01$. *** $p < .001$.

top-down scale was inversely related to several measures of cognitive flexibility and control specifically, and not related to cognitive reappraisal, suggesting it may be distinct from constructs thought to underlie cognitive-based emotion regulation strategies.

## Study 3: Relationship to experimental task

The first study demonstrated that our items reflected a two-factor bottom-up, top-down self-report measure, and our second study determined this factor structure was a statistically good fit, and that the measure related to psychopathology and some additional measures of emotion. In our third study, we sought to evaluate the BUTD measure's validity further by comparing it to performance on an experimental task. The alignment of task and self-report emotion measures is not well understood [29]. IDs and group-level experimental tasks measuring the same constructs have traditionally been pursued through separate research efforts, leading to a gap in understanding how these two approaches compare in capturing behavior. Experimental tasks of emotion are typically thought to measure states, given that they evoke real-time emotional responses. In contrast, self-report measures typically assess more stable trait constructs. One study found that self-report is more predictive of important health outcomes—such as self-reported stress [29]. This suggests trait measures might be stronger predictors of important well-being outcomes than state measures. Study 3 aimed to bridge the gap in understanding how performance on an experimental task related to self-report measures of similar constructs.

A separate, independent sample was recruited to determine whether the BUTD measure correlated with an experimental task measuring emotion generation and regulation. The task includes both top-down (e.g., cognitive reappraisal) and bottom-up components (e.g., emotional pictures). We also once again assessed the measure's fit by statistically comparing the two-factor correlated structure with the one factor structure to replicate our findings from Study 2. Finally, we assessed the BUTD measure with additional measures of positive and negative emotion and psychopathology that were distinct from the measures used in Study 2 to provide conceptual replication and better understand the shared and unique relationships of the bottom-up and top-down scales. We also included several measures (the CMBQ and ERQ) from Study 2 to determine whether the relationships would replicate in a separate sample. We predicted that both factors would be correlated with various measures from the experimental task. More specifically, we predicted that the top-down scale may relate to measures of cognitive reappraisal, while the bottom-up scale would relate to reactivity to the negative pictures. Given findings from Study 2, we also predicted both BUTD scales would be positively correlated with the individual difference measures of internalizing symptoms, emotion, and emotion regulation. We also expected that both measures, and especially the top-down measure, would positively relate to negative emotion and negatively relate to positive emotion. This study was exploratory and thus not preregistered.

## Methods

**Participants.** The data for this study was collected through a separate exploratory study that used the Prolific and SONA platforms previously mentioned in Studies 1 and 2 respectively but with new data collection separate from those two studies. Eligibility criteria included current enrollment in an undergraduate degree program at a North American university for the student SONA portion of the sample and being at least 18 years old and not older than 55 for both the student and Prolific samples.

**Ethical approval and consent to participate.** Participants completed this online study via Qualtrics between October 11th and December 23rd, 2023, and were compensated approximately $10 per hour for participation. All participants completed written informed consent via the online survey platform, and all study procedures were approved by the Institutional Review Board of the University of Denver prior to initiation of any study procedures.

The same exclusion criteria used in Study 2 were applied to the dataset to ensure quality such as exclusion based on failed attention check items included in the survey ($N=14$), and participants with scores that were greater than three standard deviations from the mean of measures collected ($N=9$). Participants were only excluded on the specific measure for which they had an outlying score based on our preregistered criteria from Study 2. This number reflects the total number of participants who were outliers on any measure. See correlation tables below for specific Ns by measure. Finally, participants were also excluded based on responses to a post-task measure which assessed their understanding of the experimental task ($N=51$). The analyses and results below are based on a final sample of 238 participants. See Table 8 for demographics information of Study 3's sample.

## Procedure

After completing informed consent, participants were trained in how to complete the online experimental task. The task was a modified version of an established, picture-based task that assesses one's ability to use cognitive reappraisal to modify emotion. This version of the task separates the generation and implementation of reappraisals. For more information on the components of the task please see [45]. For this study, the task was adapted to be completed online using the Qualtrics survey platform.

Prior to completing the actual task, participants were asked questions regarding their comprehension of the instructions. Participants could not complete the task until they had answered all comprehension questions correctly. After completing the task, participants filled out individual difference measures and demographic information (see supplemental materials for additional measures included in the study that were not central to the current study's aims). The BUTD items

**Table 8. Reported demographic characteristics of Study 3 sample.**

| Age | 29.79 +/- 9.59 |
|---|---|
| Sex | |
| Female | 115 |
| Male | 98 |
| Gender | |
| Man | 102 |
| Non-binary | 1 |
| Woman | 109 |
| Race | |
| African American | 54 |
| American Indian or Alaskan Native | 1 |
| Asian | 28 |
| White | |
| Multiracial | 11 |
| Native Hawaiian or Pacific Islander | 0 |
| Rather not answer | 7 |
| Ethnicity | |
| Hispanic/Latinx | 27 |
| Not Hispanic/Latinx | 183 |
| Rather not answer | 3 |

were presented using the same pseudo-random procedure in Study 2. Participants were compensated for their time via online monetary payment or course credit.

### Measures

In Study 3, the two subscales of the BUTD measure were scored by averaging the items. While factor scores often produce more accurate scores, factor analysis is less accessible in practice. We thus wanted to ensure our measure could be scored more simply. Correlations between the factor scores and the averaged scores for each scale were high ($r$s = +.86, $p$s < .001), suggesting the averaged scales were appropriate to use (see supplement for scoring details). The BUTD measure produced the following internal consistency metrics. Bottom-up scale: α = .71 w = .66. Top-down scale: α = .82 w = .84

### Internalizing symptom measures

**Negative and positive mood.** The Positive and Negative Affect Schedule (PANAS) was used to assess positive and negative emotions. The PANAS assesses a variety of positive and negative emotions [46]. It contains 20 items, with two subscales: 10 items assess positive emotions, and 10 items assess negative emotions. Participants were asked to rate the extent to which they have felt various emotions over the past week. Example positive and negative emotions respectively include Excited, Enthusiastic, Scared, Upset. Participants rate these emotions on a Likert scale of 1–5 where 1 is *very slightly or not at all* and 5 is *extremely*. Alpha reliability was > .91 for both subscales. Omega reliability was .91 for the Negative subscale and .92 for the Positive subscale.

**Anxiety symptoms.** Anxiety was assessed with the Generalized Anxiety Disorder screening tool (GAD-7; [47]). The questionnaire contains seven items in which participants are asked to rate the frequency of certain anxiety symptoms over the last two weeks. Sample item includes: "Feeling nervous, anxious, or on edge." Participants are asked to rate their

symptoms on a 4-point Likert scale (0 – Not at all, 3 – Nearly every day). The GAD-7 alpha and omega were.90 and.92 respectively.

**Depression symptoms.** Symptoms of depression were assessed using the Patient Health Questionnaire-9 (PHQ-9; [48]). The questionnaire contains nine items in which participants are asked to rate the frequency of certain depressive symptoms over the last two weeks. Sample items include: "Little interest or pleasure in doing things". Alpha and omega reliability were.90 and.91 respectively.

### Emotion measures

In addition, measures that were assessed in Study 2 were reevaluated in this sample including the ERQ (Reappraisal α=90, w=.91; Suppression α=.77, w=78) and the CMBQ (SR α=.92, w=.91; CM α=.92, w=.92).

### Task measures

During each of two phases (generate and implement) of the task [45], participants were asked to rate their positive emotion on a scale of 1–5, where 1 was not all positive and 5 was extremely positive. Average ratings from each of the two phases in four conditions produced eight emotions ratings, which allowed us to examine how the BUTD measure related to emotional reactivity, and the generate and implement phases of reappraisal while people had a few different goals. On control (LOOK) trials, participants did not regulate during either phase of the trial (baseline). On reappraisal trials, the first phase of each trial (GEN), participants were asked to either only generate reappraisals which increased positive emotion (+), or generate reappraisals that would increase positive emotion, *and* generate those that would increase negative emotion (+-). During the second phase of each reappraisal trial, the participants implemented (USE) only one type of reappraisal that they had previously generated (+ or -). This task has previously demonstrated that participants can successfully separate the generation of potential reappraisals from the implementation and elaboration of them, and that most of the emotional change seen in picture-based reappraisal tasks likely comes from the implementation, rather than the generation, of reappraisals [45].

### Results

**Confirmatory factor analysis.** Like in Study 2, we compared a two-model fit to a one model fit, using the same parameters described in Study 2 with the lavaan R package. The two-factor correlated model with error-covariance was the best fit (see Table 9 for model comparison and fit indices), replicating our finding in Study 2.

### Relationship to internalizing symptoms and emotion measures

We found significant relationships between the top-down and bottom-up scale and symptoms of internalizing psychopathology. We also found significant correlations with the emotion measures included. See Table 10 below.

The bivariate correlations revealed that the top-down scale was positively associated with internalizing symptoms, conceptually replicating our previous findings with different measures of internalizing symptoms. These findings further suggest the top-down scale is likely a transdiagnostic factor associated with internalizing symptoms. In contrast to Study

**Table 9. Fit indices of the two CFA models.**

| Model | $\chi^2$ | df | CFI | TLI | RMSEA | SRMR | $\Delta\chi^2$ |
|---|---|---|---|---|---|---|---|
| **1. Correlated two-factor** | **79.03** | **33** | **.94** | **.91** | **.08** | **.06** | Model 1 vs Model 2: 39.32, df diff=1, $p<.001$ |
| 2. One factor | 118.36 | 34 | .88 | .85 | .10 | .06 | |

*Note*. All models were statistically significant at the $p<.05$ level. The bolded model demonstrated the best fit of the two models.

**Table 10. Bottom-up top-down correlations with previous and new extant measures.**

|  | Top-Down | Bottom-Up |
|---|---|---|
| **Top-down Scale** | – | **.44***** |
| PHQ-9 | **.31***** | -- |
| GAD-7 | **.34**** | **.15*** |
| Cognitive-Mediation Beliefs Questionnaire – Cognitive-Mediation | -- | *.21*** |
| **Bottom-Up Scale** | **.44***** | – |
| Cognitive-Mediation Beliefs Questionnaire – Stimulus-Response | **.30***** | **.34***** |
| Emotion Regulation Questionnaire – Reappraisal | -- | *.26**** |
| Emotion Regulation Questionnaire – Suppression | -- | -- |
| PANAS – Negative Affect | **.33***** | .11+ |
| PANAS – Positive Affect | **-.16***** | .16* |

*Note*. Bolded values indicate observed value is aligned with originally predicted value. $* p < .05. ** p < .01$ $*** p < .001 += p < .1$. Italicized values indicate divergence from the relationship found in Study 2.

2 where both the bottom-up and top-down scales were positively associated with depression and anxiety symptoms, the bottom-up scale in Study 3 was only significantly correlated with anxiety symptoms. This suggests possibly a greater differentiation between the two scales and is generally more consistent with our initial hypothesis in Study 2: that the bottom-up factor would be more strongly related to anxiety than depression given its emphasis on perceptual generation of emotion.

In addition, the top-down subscale's associations with negative and positive affect, as well as the stimulus-response subscale, indicate its convergence with constructs related to emotion reactivity, generally reflecting similar relationships found in Study 2. By contrast, the bottom-up scale had a more varied relationship with emotions measures. While we replicated the bottom-up scale's relationships with the stimulus-response subscale from Study 2, we also found significant associations with measures of emotion regulation and positive affect.

### Task measures

We found significant relationships between the top-down and bottom-up scale and the task emotion ratings. Broadly, we found that the bottom-up subscale correlated with emotional ratings on the reactivity trials (non-regulation trials), and the TD subscale correlated with emotional ratings during both reactivity and regulation trials. These patterns overall suggest that the bottom-up scale was related to the unregulated responsivity to the visual elicitation of emotion, while the top-down scale was also related to emotion ratings when people were using cognition to attempt to change their emotions (Table 11). Regardless of the direction of the attempted regulation, all correlations between the top-down scale and emotion ratings were negative, such that those who reported experiencing emotions generated from the top-down reported lower positive emotions ratings to negative pictures.

## Discussion

### General summary

Overall, we successfully developed items that constituted a two-factor measure that assesses individual differences in bottom-up and top-down emotion generation. The two-factor structure provides evidence that individuals can report on emotions generated in these two somewhat separable ways. The confirmatory factor analysis also indicated that these two emotion generation processes are associated with each other. The factor analysis from Study 1 indicated a two-factor, bottom-up, top-down solution, and the confirmatory factor analyses in Study 2 and 3 indicated the two-factor

**Table 11. Bottom-up top-down correlations with components of the experimental task.**

|  | Top-Down | Bottom-Up |
|---|---|---|
| LOOK – Phase 1 | -.16* | **-.13*** |
| LOOK – Phase 2 | -.16* | **-.14*** |
| GENERATE + | **-.16*** | -- |
| USE + | -.12+ | -- |
| GENERATE + - | **-.13*** | -- |
| USE + | -.11+ | -- |
| GENERATE + - | -.11+ | -- |
| USE - | **-.21*** | -- |

*Note*. Bolded values indicate observed value is aligned with originally predicted value. * $p < .05$. ** $p < .01$ *** $p < .001$ + = $p < .1$. Each row of the table can be interpreted independently as well as in pairs to reflect the relative phases of each trial type as indicated by the line breaks (e.g., LOOK Phase 1 and Phase 2 are a pair, etc.).

solution was a good fit on two independent samples. The factors' associations with the various internalizing symptoms measures suggest both factors are associated with internalizing symptoms, and other self-report measures of emotion reactivity. More specifically, we also found that while these two scales are related, they are still separable constructs. Thus, while both scales of the measure appear to be related to emotional reactivity, the two-factor solution suggests there is utility in examining bottom-up and top-down generated emotions separately.

We also found that the BUTD measure was associated with an emotion regulation task that measures emotional reactivity and regulation. Broadly, questionnaires and task-based measures of emotional processes do not often align [29,49]. The fact that we found some evidence that our measure aligns with a well-established laboratory task lends further evidence for the possible utility of this measure in assessing relevant emotional constructs. Previous studies have examined how individual difference measures correlate with tasks and brain activity to further validate the utility of the measure in indirectly assessing important constructs [50]. In addition, we were able to conceptually replicate the BUTD measure's association with mental health symptoms using two different, well-established measures of depression and anxiety symptoms, respectively.

## Implications for affective science

The distinct and significant relationships between the bottom-up and top-down scales and the various task components in Study 3 suggest the BUTD measure may mostly reflect unregulated emotion generation. For example, given that the bottom-up measure only correlated significantly with the reactivity components of the task, without instruction to regulate, the bottom-up scale may slightly better align with how people respond spontaneously to negative stimuli, especially those driven by perception. By contrast, while the top-down scale was associated with both reactivity and regulation portions of the task, the strongest correlation was produced when cognitively upregulating negative emotions. Previous individual differences have characterized cognitive generation of emotion as highly overlapping with emotion regulation efforts to decrease negative emotion [21]. With our new measure BUTD, we report novel relationships between our top-down scale and the degree to which people were able to use cognition to increase negative emotion. Furthermore, previous experimental data indicate that neural regions involved in emotion regulation overlap with those engaged by top-down emotion generation and cognitive control, suggesting regulation success may be related to these processes [17]. However, our results indicate it is possible there is more separation between neural regions involved in cognitive control and those involved in cognitively generated emotion than previously thought. Further delineating these processes may increase our understanding of which regions or neural connections are likely to contribute to top-down reactivity versus regulation.

The BUTD measure could be used in tandem with other questionnaires of emotion regulation and experimental tasks, perhaps in tandem with neuroimaging, to differentiate between regions involved in emotion generation and regulation during a task. For example, one might predict that the bottom-up scale of our measure might relate to the response of emotional reactivity regions during unregulated responding (e.g., limbic and visual regions; [51]). Furthermore, the top-down scale of our measure might relate to the engagement of regions involved in the cognitive generation of emotion during unregulated responding or effortful up-regulation of negative emotion (mFPC; [4,5]). Finally, other individual difference measures, such as the reappraisal scale of the ERQ or the cognitively mediated beliefs scale of the CMBQ, might relate more to engagement of control-regulated regions during the regulation of emotion (dlPFC, vlPFC; [52]). Understanding these distinctions may further inform the psychological and neural mechanisms behind differences in emotion reactivity and emotion regulation success.

Furthermore, the BUTD measure will likely also be beneficial in non-experimental studies of emotion regulation. If relationships with neural regions are found, the BUTD measure could be an efficient way to capture variability in regulation and reactivity, as it is less demanding and time intensive than laboratory experiments. Given the relationships the bottom-up and top-down factors have with multiple measures of emotion regulation and internalizing symptoms, it is possible the bottom-up and top-down factors could be related to other constructs commonly related to emotion and emotion regulation, such as perceived stress, or well-being [20]. Future work should examine how the bottom-up and top-down factors relate to, and potentially mediate or moderate, other outcomes of interest.

**Implications for the study of emotion reactivity**

The relationships we found between the top-down scale, internalizing psychopathology and measures of reactivity indicate that it is important to consider that emotion reactivity variation is not just bottom-up in nature, which is often the way emotion reactivity is conceptualized in affective tasks. The top-down scale's stronger associations with constructs such as the HSP, rumination, and neuroticism in Study 2 indicate this scale may be useful in examining how strongly cognition fuels reactivity, and that in addition to examining the frequency, intensity and duration of reactivity, examining the type of reactivity is important. Ultimately these findings suggest that not all reactivity is the same in how it is derived.

While we had initially predicted the top-down scale was likely to be associated with any measure of cognition, including efforts to use cognition to regulate emotion, it appears the top-down scale is likely related to rumination (rather than constructs such as cognitive control or flexibility). While the CMBQ's Stimulus Response subscale [21] provides some measure of bottom-up reactivity, we are unaware of any previous measure that assess top-down reactivity separately from regulation, as our top-down measure seems to. As indicated by its relationship with the task data, a top-down scale of reactivity could have important implications for understanding emotional responding, including antecedents to the development of psychopathology. In our task, we also found that top-down scores appeared to be most related to effective upregulation of negative emotions. The separation of positive and negative reappraisals in our task allowed us to more strongly delineate that our measure characterizes how cognition fuels negative emotions, despite cognition's general reputation as diminishing negativity. In the context of our task, it appears those with higher top-down scores are likely to experience lower levels of positive emotions. If this generalizes to other contexts, it likely has implications for the development and maintenance of psychopathology.

We initially predicted the bottom-up scale would relate to measures of perceptual reactivity. While we did find some evidence for this, we also found the bottom-up scale positively related to several measures of emotion regulation in Studies 2 and 3. One possibility is focusing on the concrete environmental aspects of the situation may increase the likelihood of engaging in strategies such as reappraisal. Reappraisal strategies often encourage people to focus on objective forms of evidence relative to the initial appraisal to help challenge unhelpful thinking patterns [53] and ultimately reduce or change negative emotions. These results need to be replicated and extended to fully understand this relationship.

## Clinical relevance

In recent years, clinical research has shifted to examining transdiagnostic risk factors, or shared underlying/vulnerability processes across psychological disorders rather than presuming that the mechanisms underlying each disorder are distinct from all other disorders. This shift in focus from individual diagnostic groupings to a transdiagnostic approach, largely prompted by the National Institute of Mental Health's Research Domain Criteria (RDoC), seeks to identify mechanisms that may underlie the development and maintenance of psychopathology across disorder groupings. Identifying these mechanisms will improve our ability to treat these problems more directly and more efficiently by targeting the underlying processes themselves [54]. For example, historically internalizing disorders such as anxiety and depression have been studied separately, despite shared overlap in emotional processes (e.g., worry and rumination) and behavior (e.g., avoidance and withdrawal). However, in the last two decades, researchers have begun to identify emotional processes core to both disorders. One such transdiagnostic factor is emotion regulation, which is now considered to be a known risk factor across both anxiety and depression (and a variety of other disorders; [55,56]).

While clinical research has largely focused on examining emotion regulation as a mechanism underlying psychopathology, less emphasis has been put on the importance of bottom-up and top-down emotion generation, despite a theoretical rationale that nearly all psychological disorders, and especially the internalizing domains (anxiety and depression) include in the core diagnostic criteria unhelpful generated emotions (e.g., irrational fear responses or sadness/irritability that drive impairing behavior; avoidance or withdrawal; [57]). Emotion generation is a critical process to study in the context of mental health symptoms given that emotion regulation occurs in the context of experiencing and subsequently regulating emotions like fear and sadness. However, the literature has not yet fully explored individual variation in bottom-up versus top-down emotions and how this might be related to internalizing symptoms. For example, anxiety disorders involve heightened perceptual awareness of threatening stimuli [58], which may be driven by more external sources of emotions, which are likely to be bottom-up processes, while depressive disorders involve ruminative thinking [32,59], which may be driven by more internal sources of emotions, and thus might be more top-down driven. A wealth of research across anxiety and depression has examined both internal and external processes that likely contribute to psychopathology symptoms,that are present in depression [60] and anxiety ([61], for a review see [62]), many of which overlap across the disorders [63]. Given the difference in internal or endogenous versus external or exogenous sources of emotion, it is possible these may be driven by differences in bottom-up and top-down emotion generation. Ultimately, the literature suggests that anxiety and depression both share biases toward negative emotional stimuli, but the source (internal versus external) may differ between the disorders (more internal for depression, and more external for anxiety [63].

The BUTD measure demonstrated consistent, modest associations with internalizing symptoms, suggesting it could be used to better understand clinical outcomes. In particular, the top-down factor had consistent significant correlations with both anxiety and depressive symptoms. The bottom-up scale had consistent significant correlations with anxiety, but less reliable relationships with depression. Overall, these findings suggest that both scales likely relate to depression and anxiety symptoms, but it is possible that the top-down scale captures more of the shared process between the two disorders, and that the bottom-up scale measure may relate more to anxiety, and thus somewhat differentiate between internalizing symptoms.

Anxiety and depression are highly comorbid, and there is increasing evidence of heterotypic continuity – that the same mechanisms underlying these processes are simply phenotypically expressed more as fear and worry (in the case of anxiety) and/or as low mood, irritability, and/or anhedonia (as in the case of depression; [64,65]). The strong relationship between these two mental health domains aligns with the RDoC conceptualization. RDoC currently includes cognitive and emotional processes that may be linked to bottom-up (e.g., attention bias for threatening faces) or top-down (interpretation of ambiguous information as threatening or negative) generated emotions. However, RDoC does not currently differentiate top-down and bottom-up processes, and it remains to be seen whether they are the same or different as they relate to different forms of psychopathology.

## Clinical application

The BUTD's efficiency could also lend well to aiding in the development of treatment targets, as there's often not time or understanding of how to use experimental tasks to inform clinical treatment. If bottom-up and top-down emotion generation variability is related to underlying processes of psychopathology, the BUTD measure might also help identify specific treatment targets for intervention, in addition to specific emotion regulation strategies most likely to be effective given an individual's unique endorsement of bottom-up and top-down emotions. For example, it could be that individuals who report elevated top-down emotion generation and also experience depression and anxiety may more readily engage in appraising situations as negative or threatening (e.g., appraisal bias; [66] and thus may be more amenable to cognitive based therapy strategies intended to alter negative appraisals such as CBT [67,68]. In comparison, individuals who report elevated bottom-up generated emotions and also experience anxiety may demonstrate heightened awareness to environmental cues. In this case, it may be that a behavioral approach that targets reactivity to those environmental stimuli, such as graded exposure, could be more successful in reducing symptoms [69]. Importantly, a 10-item measure that may have direct implications for teaching individuals strategies to regulate emotions has high potential for clinical utility with low burden on the participant/client. Recommended next steps include examining how the BUTD measure may relate to different symptoms of depression and or anxiety over time, including how they change over the course of treatment, and how the BUTD measure relates to different components of treatment.

## Limitations

This three-part investigation had many strengths, including a sequence of exploratory and confirmatory analyses in separate samples, pre-registration of our methods and hypotheses for confirmatory analyses, the use variety of self-report measures, and an experimental task to examine the measure's validity. Many of the established self-report measures we selected for our studies were included in other measure development studies, including the evaluation of the ERQ and CMBQ. This ensured we included an overlapping set of relevant affective constructs aligned with previous measure development. As with every study, there are also some limitations to keep in mind. The current reported studies were predominantly composed of young adults who identified as White cisgender women with access to higher education, thus limiting generalizability. However, our inclusion criteria in Study 3 were broader, which allowed us to examine the measure's fit and validity in a more diverse sample. Notably, we used different measures of clinical symptoms, in an attempt to conceptually replicate findings. More specifically, we used both brief and more comprehensive measures of anxiety and depression. The PHQ-9 and GAD-7 are often used as brief screening tools whereas the BDI and STAI are more comprehensive in their assessment of various symptoms. These differences may have contributed to variability in relationships across our studies but also strengthened our interpretation of findings that are consistent across measures. Future work could include measures that detect symptoms of depression in non-clinical samples to better assess the BUTD measure's relationship with such instruments.

In addition, the correlations with existing measures were only partially in line with our predictions, and some varied by study. For example, the bottom-up factor correlated with several cognitive measures in Study 3, and the top-down measure did not, both of which were counter to predictions. In addition, the bottom-up factor was predicted to be positively related to measures of environmental sensitivity, which we observed, but so was the top-down factor (which was not predicted). Therefore, while the two-factor model supported that our bottom-up and top-down scales measured somewhat separable constructs, there was mixed evidence to support the factor scores had distinct relationships with other measures studied. The results suggest that both factors, and particularly the top-down sub-scale, were likely capturing aspects of emotional reactivity more broadly, although the two-factor structure does indicate that they were not measuring redundant constructs.

Differences across our studies may reflect the differences in our samples across studies. Given that Study 3 was our most diverse sample and produced the strongest overall reliability for the BUTD scale, these findings may be considered

more interpretable than Study 2, for which the sample was less generalizable. Because of the well-established task included in Study 3, we were able to more comprehensively examine the measure's utility in relation to other methods commonly used in affective science. However, both the task and BUTD measure rely on self-report, which requires a certain level of emotional awareness to be useful. A future study using more objective forms of emotion measurement may be helpful. Providing options for other-informant responses may also be advantageous for understanding how BUTD emotional responding may be perceived or observed by others. Additionally, Study 3 was exploratory, and therefore the correlations we found with the task should be considered preliminary until future work can replicate our findings.

Finally, the bottom-up sub-scale had relatively poor alpha reliability in Study 2, however its omega score indicates acceptable reliability in all studies. In addition, the bottom-up measure's reliability across both alpha and omega scores was acceptable in Studies 1 and 3. Given this discrepancy, Study 2 results should be interpreted with more caution as this may indicate potential instability of the scale. This may also reflect an effect of age, given that the average age in Study 3 was 10 years older than in Study 2. It may be that reflecting and reporting on bottom-up generated emotions is more challenging for young adults. Differences by age in emotion research have been found previously, suggesting that as people age, they report higher rates of well-being [70] and improved ability to regulate emotions [70,71]. It is possible a similar mechanism may be taking place for the bottom-up scale. The top-down factor, however, did have adequate reliability in both studies, suggesting a stable and consistent factor to use in future research, and potentially less susceptible to differences across samples.

Future studies should address these limitations and expand on the current work by continuing to examine how the BUTD measure may differ across more heterogenous, and representative, samples. In addition, examining the measure's predictive validity, its relationship to the p factor [72], psychological flexibility [73], and other measures of internalizing and externalizing psychopathology will further clarify the measure's utility. Lastly, there are numerous individual difference measures examining varied aspects of emotional responding. It is of course possible there are other measures that should be included in future work examining bottom-up and top-down generated emotions. For example, the difficulties in emotion regulation difficulties scale [74] which assess one's ability to understand and regulate their emotions may provide additional information regarding how emotional awareness and difficulties regulating emotions relates to bottom-up and top-down reactivity. In addition, measures of irritability, well-being, emotional awareness, and broader cognitive processes have been used in other studies examining emotional responding [21,20] and thus may provide additional discriminant and concurrent validity.

## Future directions

In addition to the above-mentioned future directions, conducting measurement invariance analyses using an adequate sample of respondents with various genders and ages would be an important next step for further examining the psychometric properties of the BUTD measure. It may also be of benefit to expand this model in future work by using a wider array of psychological attributes to examine other models, such as hierarchical or bi-factor structures. Given there is evidence supporting gender differences in mental health symptoms [75,76] and emotion regulation [77,78], it is possible that similar patterns may emerge in beliefs about emotion generation as well (note: these studies did not explicitly examine differences in individuals who identify outside of the gender binary). Indeed, a recent study suggests there are likely meaningful gender differences in emotion recognition [79], such that men appear to have a bias towards male threat stimuli that women do not exhibit.

The stability of these individual differences over time, including the measure's test-retest reliability, should be examined in future work to further address the utility and psychometric properties of the measure. It would also be of benefit to examine multi-level mechanisms that underlie the development of variability in bottom-up and top-down emotions. For example, the etiology of relatively greater cognitive (versus environmental) representations of emotions than others is unclear. It is possible that societal or cultural factors could influence this development. Cultural differences in the expression of emotion

influence mental representation of emotions including perception of emotional intensity [80]. It is possible that behavioral practices and social expectations in certain cultures can subsequently influence one's sensitivity to bottom-up emotions as they attend to certain perceptual features. Future research should work to elucidate how different cultural factors could affect bottom-up emotion generation as well as top-down emotion generation. Finally, future work should consider collecting relevant clinical information on the sample including use of substances, prescribed medications, and mental health history.

## Conclusion

We determined that an individual difference measure of bottom-up and top-down emotions could be developed and assessed the measure's factor structure. Its reliability and validity were evaluated in three independent samples. Exploratory analysis indicated, and confirmatory analyses confirmed, a two-factor solution with two separate factors for bottom-up and top-down emotions, supporting the idea that bottom-up and top-down emotions can be measured via self-reported questionnaires, when they have previously only been measured via experimental tasks. Both scales demonstrated adequate internal consistency. The correlation analyses revealed that the top-down factor was consistently associated with symptoms of depression and anxiety, and that the bottom-up factor was consistently associated with anxiety, suggesting a transdiagnostic relationship between top-down emotions and internalizing symptoms, and possibly a specific relationship between bottom-up emotions and anxiety. Relationships between the scales and extant measures in the second study only partially aligned with the initial hypotheses; the top-down factor broadly did not relate to the measures of cognition, and the two factors did not yield strongly differential relationships across different measures, suggesting limited discriminant validity. In contrast, relationships between the scales and extant measures in the third study better aligned with our initial hypotheses; that the top-down scale would relate to measures of cognitively generated emotions across task and questionnaires, while the bottom-up measure related to bottom-up generated emotions.

Future work should examine how the BUTD measure relates to psychophysiological and neural measures of bottom-up and top-down emotions, as well as emotion regulation. Relatedly, assessing reaction time during a task and its association with the BUTD measure may provide additional insight into variability in processing speed of different emotions. Variability in reaction time could be important from an evolutionary standpoint (e.g., faster reaction time to threatening stimuli). Additional research will be needed to determine how the BUTD measure may apply to other factors of interest including mental health, cultural context, and gender identity, which could influence individual differences in emotion generation.

## Supporting information

**S1 Table. Predicted correlations between the new and extant measures. primary and secondary measures of interest and their relation to bottom-up and top-down emotions.** Correlation strengths are defined as: weak $r$=.1-3, moderate $r$=.3-.5, strong $r$=.5-.8.
(DOCX)

**S2 Table. Factor Loadings of the CFA.**
(DOCX)

**S3 Table. Comparison of the CMBQ and BUTD measure.**
(DOCX)

## Acknowledgments

We'd like to thank Angela Narayan, PhD for her insightful feedback throughout the duration of this project. We'd also like to thank Christian Waugh and Valeriia Vlasenko for sharing their experimental task with us, and Silver Martin and Sophie Rosenblatt for their data collection efforts.

## Author contributions

**Conceptualization:** Nadia Kako, Kateri McRae.

**Formal analysis:** Nadia Kako.

**Funding acquisition:** Kateri McRae.

**Methodology:** Nadia Kako, Denis Dumas, Kateri McRae.

**Project administration:** Nadia Kako.

**Supervision:** Denis Dumas, Kateri McRae.

**Writing – original draft:** Nadia Kako, Michelle Rozenman, Kateri McRae.

**Writing – review & editing:** Nadia Kako, Michelle Rozenman, Denis Dumas, Kateri McRae.

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
