## [Decision Letter · Decision Letter 0]

5 Aug 2025

PMEN-D-25-00274

Individual differences in bottom-up and top-down emotion generation

PLOS Mental Health

Dear Dr. Kako,

Thank you for submitting your manuscript to PLOS Mental Health. After careful consideration, we feel that it has merit but does not fully meet PLOS Mental Health’s publication criteria as it currently stands. Therefore, we invite you to submit a revised version of the manuscript that addresses the points raised during the review process.

The reviewers have outlined some points that will improve the quality of the manuscript, including positioning it more clearly within the literature and outlining the potential impact for the field. Addressing these comments is likely to result in an improved manuscript. Please consider them carefully and submit a revised manuscript, outlining which edits and changes have been made to address the reviewers' comments.

We look forward to receiving your revised manuscript.

Kind regards,

Randall Waechter

Academic Editor

PLOS Mental Health

Journal Requirements:

1. Please provide a/amend your detailed Financial Disclosure statement. This is published with the article. It must therefore be completed in full sentences and contain the exact wording you wish to be published.

1. Please clarify all sources of funding (financial or material support) for your study. List the grants (with grant number) or organizations (with url) that supported your study, including funding received from your institution.

2. State the initials, alongside each funding source, of each author to receive each grant.

3. State what role the funders took in the study. If the funders had no role in your study, please state: “The funders had no role in study design, data collection and analysis, decision to publish, or preparation of the manuscript.”

4. If any authors received a salary from any of your funders, please state which authors and which funders.

2. We have amended your Competing Interest statement to comply with journal style. We kindly ask that you double check the statement and let us know if anything is incorrect.

3. Please note that your Data Availability Statement is currently missing the repository name and/or the DOI/accession number of each dataset OR a direct link to access each database. If your manuscript is accepted for publication, you will be asked to provide these details on a very short timeline. We therefore suggest that you provide this information now, though we will not hold up the peer review process if you are unable.

4. Please provide separate figure files in .tif or .eps format.

https://journals.plos.org/mentalhealth/s/figures

https://journals.plos.org/mentalhealth/s/figures#loc-file-requirements

5. Please upload a copy of Figure 1 which you refer to in your text on page 13. Or, if the figure is no longer to be included as part of the submission please remove all reference to it within the text.

6. We notice that your supplementary tables are included in the manuscript file. Please remove them and upload them with the file type 'Supporting Information'. Please ensure that each Supporting Information file has a legend listed in the manuscript after the references list.

Additional Editor Comments (if provided):

Reviewers' comments:

Reviewer's Responses to Questions

**Comments to the Author**

1. Does this manuscript meet PLOS Mental Health’s publication criteria?

Reviewer #1: Yes

Reviewer #2: Yes

Reviewer #3: Yes

2. Has the statistical analysis been performed appropriately and rigorously?

Reviewer #1: Yes

Reviewer #2: Yes

Reviewer #3: Yes

3. Have the authors made all data underlying the findings in their manuscript fully available (please refer to the Data Availability Statement at the start of the manuscript PDF file)?

Reviewer #1: Yes

Reviewer #2: No

Reviewer #3: Yes

4. Is the manuscript presented in an intelligible fashion and written in standard English?

Reviewer #1: Yes

Reviewer #2: Yes

Reviewer #3: Yes

Reviewer #1: This is well-written and clear.

Abstract doesn't contain any numeric detail (sample size, results)

Introduction seems to neglect personality traits - not true that 'less has been done to capture individual variation'. There is a large literature on personality traits, individual differences and emotion.

Structure is odd - far too early to sat 'We sought to determine' - research questions should go at the end of an introduction.

Later, again the suggestion is made that 'less emphasis has been put on the importance of emotion generation'. There is a vast literature on this already.

There should be a 'Methods' section containing Study 1/2/3, rather than 'Study 1' being the first heading.

It is surprising to use Prolific for Study 1, then a student sample for the confirmatory part. I would have done this the other way around, so that the general population sample is the validation sample. What is the rationale for this?

Table 6 - at first glance I thought this was a one-factor model due to layout. Can you spread across two columns for clarity.

Table 7 - despite the two-factor structure, the pattern of correlations with validating scales suggests a general factor of negative emotionality.

Study 3 - this was not pre-registered but there was a clear research question. Why couldn't that be pre-registered?

Proofreading is needed throughout - several references are missing square brackets and the number is not formatted correctly in the text. L849 has "=" for example. Missing reference at L882.

Was there an opportunity to record reaction time during Study 3 tasks? Might reaction time offer insights into speed of processing of top down / bottom up emotion generation?

Does speed of processing of information in the environment relevant to emotion impact emotion, so it becomes a cognitive skill of sorts? Is there a possible evolved mechanisms for this that increases survival e.g. threat perception being accurate and fast?

The discussion section should be rewritten to consider more closely strengths and weaknesses in relation to existing studies, to address concerns that these constructs are jangles for existing constructs including neuroticism. Limitations of self vs informant reports need acknowledging.

There is some discussion of implications for understanding psychopathology, but the scales used to validate your constructs focused on normal rather than abnormal constructs. At L788 - without stepping too far beyond the data, what are the possible implications?

Avoid "it's" throughout, which sounds informal ("it is").

Add to limitations, predictive validity for (longitudinally measured) outcomes lacking (e.g. health outcomes). Relationship to internalising/externalising psychopathology needs further study, and the "p" factor.

Reviewer #2: This paper has been written fairly well. However, I would like to suggest a few points that would make the paper more eligible for publication. There is no scientific justification for the need of this research and what is the added value of this findings to the medical/scientific fraternity. It is imperative to include the location of research, sample selection criteria, inclusion and exclusion criteria and method of data collection. The discussion section should include more studies that support or refute your findings. More references are needed from the recent studies conducted.

Reviewer #3: This is an interesting and important set of data that has implications for the treatment of emotion processing disorders. Whilst the validation of the data has been conducted with relative rigour the context of the research question could benefit from a more detailed theoretical framework.

It would benefit from a more nuanced and detailed examination of the foundations for the research question.

I would like to note that I enjoyed reviewing this manuscript.

Introduction

Whilst there are supporting citations regrading “top-down” and “bottom-up” it might be helpful to add a little more context. Especially as the terms originated in work in visual perception in the late ‘60’s and into the 1970’s. I am not suggesting that these origins need to be detailed just perhaps a little more context in terms of the exogenous and endogenous aspects to clarify the terms in the context of emotion. Whilst this is expanded upon later in the introduction, I am not convinced that this adequately address the complexity of the relationship between a stimulus driven response and a cognitive or biologically driven response. The relationship between the two is perhaps oversimplified here?

Lines 50.51: I am not sure that its clear what is meant here in relation to individual differences and group differences in relation to emotion in relation to the above. This could be expanded and made more explicit.

Lines 83-96:

I think its important to distinguish between the structural accounts of emotion processing in relation to TD-BU and the temporal accounts in imaging studies in ERP/EEG studies which might help shed some light on these concepts.

The section on psychopathology would benefit from a more detailed account. There is actually quite a large literature on these issues that would support a stronger and more balanced argument about how endogenous and exogenous accounts relate to TD BU accounts in the psychopathology of emotion processing disorders.

Methods

Line 184: This is a good example of where the explicit foundations for the research question are implicit in the supporting literature raised in the introduction. I would suggest working backwards to expand support for this claim in a more detailed examination of the literature that supports this claim.

The co-production aspect of the study is a strong positive point.

Line 210: I am assuming this is a typo? Dat/Data?

I would argue there needs to be a stronger argument for use of the Beck over other measures like the CES-D. It might be helpful to discuss the role of self-report measures in the context of emotion processing. Was there support offered for those participants who scored highly on the Beck for example? Was there any data collected in regard to participants who might be prescribed anti-depressants? Or have a history of being prescribed as research is clearly supporting an extremely long withdrawal in terms of years rather than weeks to SSRI’s in particular? Some data suggests levels at 60% in undergraduate populations.

Was the criteria for participation only age? Was there any consideration of other demographic and psychological variables? Emotion processing disorders, screened or self-reported for example?

Results

The treatment of data is reported well. However there are some interesting interpretations that need unpacking and again, perhaps highlight some of the oversimplistic theoretical framework that drives the research. For example the introduction of the role of threat accounting for BU responses.

Line 511: It would be interesting to see if this was any different using a measure of Cognitive fusion rather than psychological flexibility? I would be interesting to examine this in light of ACT therapy.

Line 623: there seems to be an error in citation here?

PQH-9 & GAD-7 as diagnostic measures are very different in interpretation that the Beck and STAI for example? Maybe this should be expanded on in more detail?

Discussion

In the general discussion the results presented in relation to structural accounts of emotion processing are a little tenuous. I would temper those. The measure developed in relevant and interesting but to make those claims support from imaging data would be necessary.

Whilst the authors note the limitations of their sampling there are other limitations that might make interpreting the data difficult. More detailed information in regard to the sample would be helpful? Recreational drug use, prescribing history etc.

In sum this is a useful measure that might help better understand the role of two potentially distinct models of emotion processing but more work is needed to fully support these claims.

**Do you want your identity to be public for this peer review?** For information about this choice, including consent withdrawal, please see our Privacy Policy

Reviewer #1: **Yes:** Gareth Hagger-Johnson

Reviewer #2: No

Reviewer #3: No

---

## [Editor Report · Decision Letter 1]

3 Dec 2025

Individual differences in bottom-up and top-down emotion generation

PMEN-D-25-00274R1

Dear Mx. Kako,

We are pleased to inform you that your manuscript 'Individual differences in bottom-up and top-down emotion generation' has been provisionally accepted for publication in PLOS Mental Health.

Best regards,

Randall Waechter

Academic Editor

PLOS Mental Health